# Recyclable photoresins for light-mediated additive manufacturing towards Loop 3D printing

Xabier Lopez de Pariza [1], Oihane Varela[1], Samantha O. Catt [2,3], Timothy E. Long[4], Eva Blasco [2,3] & Haritz Sardon [1] ✉

Additive manufacturing (AM) of polymeric materials enables the manufacturing of complex structures for a wide range of applications. Among AM methods vat photopolymerization (VP) is desired owing to improved efficiency, excellent surface finish, and printing resolution at the micron-scale. Nevertheless, the major portion of resins available for VP are based on systems with limited or negligible recyclability. Here, we describe an approach that enables the printing of a resin that is amenable to re-printing with retained properties and appearance. To that end, we take advantage of the potential of polythiourethane chemistry, which not only permits the click reaction between polythiols and polyisocyanates in the presence of organic bases, allowing a fast-printing process but also chemical recycling, reshaping, and reparation of the printed structures, paving the way toward the development of truly sustainable recyclable photoprintable resins. We demonstrate that this closed-loop 3D printing process is feasible both at the macroscale and microscale via DLP or DLW, respectively.

Additive manufacturing (AM), also known as 3D printing, emerged as a rapidly developing manufacturing method with a high impact on industrial processing[1,2]. A myriad of complex 3D structures from microscopic[3] to macroscopic[4] scale have been reported, with applications ranging from general purpose objects to engineering materials or smart devices performing in the boundary of electronic and biological systems[5–11]. Advances in computing, engineering, and processing methods rely on chemistries which enable the process to be feasible, and more recently aim at unattainable properties such as recyclability or performance[12]. Among AM methods vat photopolymerization (VP) techniques are well established and considered one of the advanced AM techniques owing to the improved efficiency and printing resolution at the macroscale[13,14]. VP processes rely on the cross-linking reaction of photopolymerizable resins under light irradiation which

enables the gelation of a layer with stereo control (2D shape) that can be differentiated from the non-irradiated liquid resin[15,16]. By continuous repetition of this process (layer-by-layer) 3D objects are produced with high resolution, leading to this technique being one of the most relevant 3D printing methods for diverse applications from soft-tissues to high modulus glasses[17,18]. Recent advances in the field permitted volumetric printing of photopolymers yielding completely freeform 3D structures[3,19,20]. Moreover, two-photon direct laser writing (DLW) allowed the printing of micro- and nanoscale features by irradiation of a photosensitive resin using multiphoton absorption exhibiting an outstanding resolution of the printed structures[3,21].

While intensive research has been focused on designing printing methods to improve printing efficiency, allow multi-material printing processes or enhance the resolution, still light-based 3D printing

[1]POLYMAT and Department of Polymers and Advanced Materials: Physics, Chemistry and Technology, Faculty of Chemistry, University of the Basque Country UPV/EHU, Donostia-San Sebastián 20018, Spain. [2]Heidelberg University, Institute for Molecular Systems Engineering and Advanced Materials (IMSEAM), 69120 Heidelberg, Germany. [3]Heidelberg University, Organic Chemistry Institute (OCI), 69120 Heidelberg, Germany. [4]Arizona State University, School of Molecular Science and Biodesign Center for Sustainable Macromolecular Materials and Manufacturing, Tempe, AZ 85281, USA. ✉e-mail: haritz.sardon@ehu.es

methods are almost solely limited to non-recyclable free radical or cationic photopolymerization of (meth)acrylate or epoxide monomers, constraining the range of photoprintable formulations[10]. Indeed, the cross-linked nature of photoprinted objects provides higher chemical resistance and superior high performance properties but does not allow them to be reprocessed or recycled, leading to the generation of a vast amount of waste, implying strong environmental concerns[12,22].

One way to shift from the aforementioned linear economy process (use and discard system) to a more sustainable circular economy model would be to incorporate dynamic covalent bonds (DCBs) into the photopolymer formulation[12,23]. In this regard, cross-linked photopolymers that can maintain their original properties under in use conditions while, when a specific external stimulus is applied (e.g., heat, light irradiation), the cross-linked nature of the material is reduced due to the activation of the DCBs are desired[24,25]. This approach has been previously used in literature to recycle/reprocess cross-linked photopolymers[26]. Bowman and coworkers showed the use of photocross-linked thiol-ene[27] based resins bearing dynamic thioester bonds which degrade into thiol-terminated oligomers by the thiol-thioester exchange. These were repolymerized into materials with identical properties over three cycles. Jin and coworkers also recently reported a strategy based on commercially available monomers using the thiol-disulfide exchange to chemically recycle cross-linked thiol-ene photopolymers[28]. An obvious advantage of designing these photocross-linked polymers is to circumvent the burden of developing resins with limited recyclability.

DCBs in VP has been explored in recent years due to the versatility offered by dynamic bonds in the printed structure. In this regard, printed objects with reshaping, self-healing and/or reprocessing possibilities are reported in recent reviews[29,30]. The impact of DCBs in VP has significantly received less attention for reprintability possibilities of 3D printable materials. Zhao and coworkers reported a reprintable system based on the photopolymerization of a monofunctional acrylate to form linear polymers, which rapidly transform from liquid to 3D objects of linear polymers with high resolution[31]. The printed polymers were subsequently dissolvable in the neat monomer to afford a reprintable resin. This approach addresses the recyclable limitations of VP printed thermosets although provides inferior high performance properties in terms of solvent resistance or thermo-mechanical property of thermosets (Fig. 1a). Wang and coworkers also recently reported a 3D printable recyclable two-stage curing thermoset system composed by photoactive acrylate and epoxy hybrid resin[32]. The proposed mechanism follows traditional acrylate photocuring followed by a thermal treatment to initiate the epoxy polymerization together with a partial copolymerization of the epoxy-acrylate system. The subsequent thermosets were depolymerized using excess ethylene glycol to promote depolymerization via transesterification to afford a viscous liquid residue. However, the depolymerization residue was reformulated with fresh photopolymerizable resin to afford printable objects as no photoactive group remained available after the depolymerization process. Thus, the depolymerized oligomers did not bear photocurable groups and therefore, new acrylate sites were added after each cycle to perform the photopolymerization (Fig. 1b). Despite this emerging attention in the literature, the development of a more sustainable composition together with a sustainable process remains as an attractive opportunity for investigation.

Inspired from earlier literature using DCBs, our research delivers photocurable resins with circularity built into their performance using commercially available monomers. We envisioned that to afford a more circular approach for the preparation of photoprintable resins with retained properties, the DCB allowing the depolymerization must be the bond generated during the printing process. This unprecedented process allows to produce resin formulations and therefore photocured thermosets with retained properties to the pristine materials. Thus, conventional acrylate photopolymerization strategies are not suitable due to the non-dynamic nature of the formed C–C bonds of the thermoset backbone and therefore additional photo-triggered strategies are required for the printing process (Fig. 1c).

Besides being recyclable these resins must allow a (1) fast photocuring process, (2) facilitate large-scale prototyping by using industrially accessible monomers, and (3) the preparation of materials of different properties to potentially be used in a myriad of applications. In this regard, poly(thio)urethane (PSU) networks have been shown to easily depolymerize into liquid oligomers under basic conditions, which were subject to re-polymerization into indistinguishable materials[33]. The formulation design for PSU thermosets relies on the click-reaction of thiols (R-SH) with isocyanates (R′-NCO) to yield thiourethane (R-S-CO-NH-R′) linkages in the backbone through a base catalyzed step-growth polymerization process[34]. This approach has been investigated using latent organic bases activated either by light[35,36] or thermally[37] enhancing the control over the curing reaction. Besides, PSUs are a very versatile polymeric material with excellent mechanical[38], physical[39], and optical[40] properties as well as biocompatibility[41,42] although they have been investigated to a lesser extent than polyurethane counterpart materials[43]. We hypothesize that through the selection of an appropriate photobase generator (PBG), thiourethane-based resins will be ideal candidates for the closed-loop 3D printing process[44]. This unprecedented approach demands a unique convergence of polymerization processes and organic catalysis together with the required rheology and photo-reactivity required for efficient vat photopolymerization.

Herein, in our efforts to transform the VP method to a more sustainable recyclable circular process, we exploited the temporal- and stereo-control offered by PBGs in the base catalyzed thiol-isocyanate click-reaction for the production of poly(thio)urethane 3D objects. PBGs have received significantly less attention to free radical counterparts, but further fundamental understanding will provide much needed diversity to the range of polymeric precursors for vat photopolymerization. Moreover, using commercially available monomers, the described photocurable resin shows feasibility for printing in the macroscale or microscale via DLP or DLW, respectively. The unique application of the dynamic character of thiourethane bonds provides 3D objects that can be repaired but also chemically recycled into thiol-terminated oligomers, which can undergo closed-loop into 3D structures with identical properties and appearance as the initial forms, after with the addition of fresh isocyanates, thus substantially increasing the sustainability of the 3D printing process.

## Results and discussion
### Resin formulation and photocuring
As an initial proof of principle, we studied a liquid resin (PSU1) composed of equimolar amounts of isocyanate terminated prepolymer and a trifunctional thiol as cross-linker that would polymerize into poly(thio)urethane thermosets through base-catalyzed step-growth polymerization. As a latent photobase generator we used a tetraphenylborate salt derivate together with a visible light sensitizer (refer to "Methods"). Safety equipment must be used due to the use of thiols and isocyanate terminated prepolymer, which have shown some health concerns. The sensitizer was used to photochemically release the free base from the tetraphenylborate salt under UV-A (365–385 nm) light irradiation through triplet energy transfer from the sensitizer to the tetraphenylborate salt[45,46]. In order to solubilize the catalyst system into the monomer mixture and reduce the viscosity of the resin, acetone (30 wt%) was used (Fig. 2a, b).

One key factor for the preparation of resin formulations for VP is their dark stability and photorheological behavior. In fact, resins which are not stable (remain in the liquid form) during the printing process will not be suitable for VP[15]. Besides, a sharp increase of the storage modulus is required when irradiating the resin to be able to obtain

well-defined objects during the 3D printing process[47]. The dark stability of PSU1 was confirmed by tracking the evolution of the storage modulus (G′) and loss modulus (G″) over 2.5 h. No cross-linking of the material was observed within this time with a slight increase in the G″ due to solvent evaporation, verifying the stability of the resin (Fig. 2d). A control experiment performed without the catalyst system further confirmed the suitability of the formulation. Indeed, in spite of the inherent reactivity of isocyanates these formulations could be stored at low temperature and dry environments up to a month without suffering any change in the resin. However, when PSU1 was irradiated with a UV-LED light source centered at 365 nm with an intensity of 20 mW/cm$^2$ a substantial increase of the moduli was observed with a

crossover point (solid-like behavior) within 45 s of irradiation (Fig. 2e). A sharp photo-hardening profile is crucial for efficient VP process. By the choice of the TBD·HBPh$_4$:ITX photocatalyst system, this was achieved under mild UV-A irradiation intensities, which are required for its application in commercially available VP printers. The correct formation of the PSU network was confirmed by FTIR spectroscopy, which showed the consumption of the isocyanate band at 2250 cm$^{-1}$ and the subsequent appearance of the thiourethane band at 1750 cm$^{-1}$ (Fig. 2f)[43]. Then the catalyst:sensitizer ratio was optimized. It was observed that both the PBG and the ITX sensitizer were required for the activation of the polymerization, with a 2:1 TBD·HBPh$_4$:ITX weight ratio being the most efficient (Supplementary Fig 1a). Increasing the

Previous works

**a** 3D printing of reprintable thermoplastics

**b** Partial depolymerization acrylate-epoxy doble networks

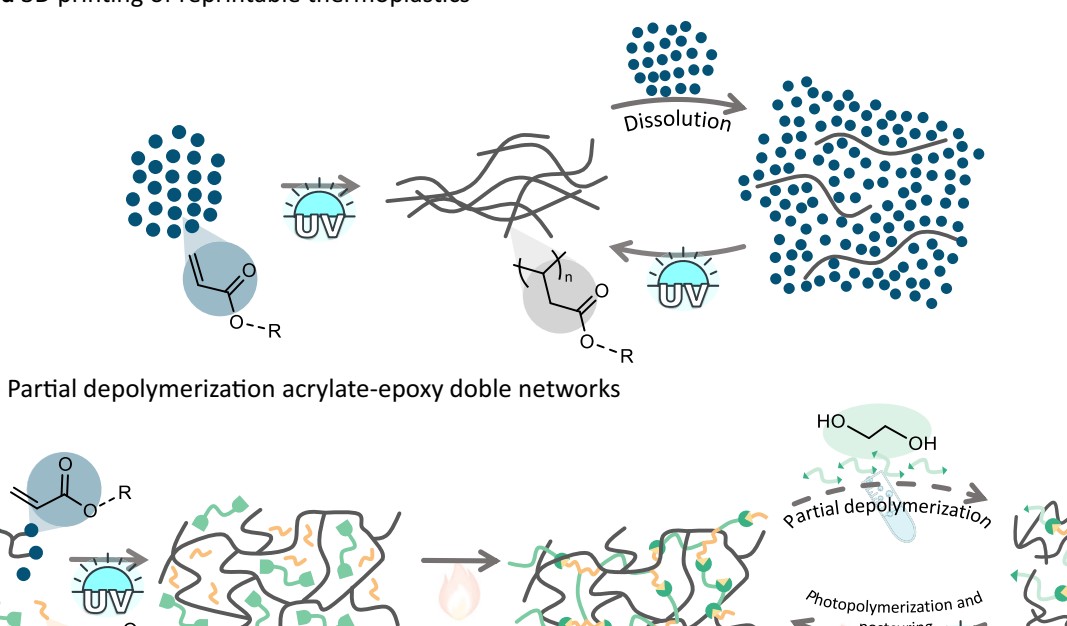

Current work

**c** Direct photocuring of dynamic thiourethane bonds

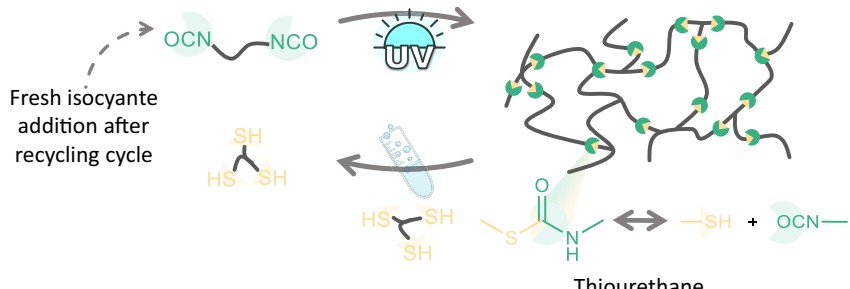

**Fig. 1 | Recyclable 3D printable systems. a** Photoprinting of linear thermoplastics, which are soluble in the monomer. **b** Photoprinting of dual systems containing photocurable monomers and heat curable monomers. Further depolymerization by dynamic covalent bond exchange to yield liquid residue and subsequent photocuring by addition of fresh resin. **c** Loop-printing process where DCB are printed using photobase generators, which are further depolymerized using an excess of initial thiol monomer to yield functional oligomers which are reprintable by addition of isocyanate functional monomer.

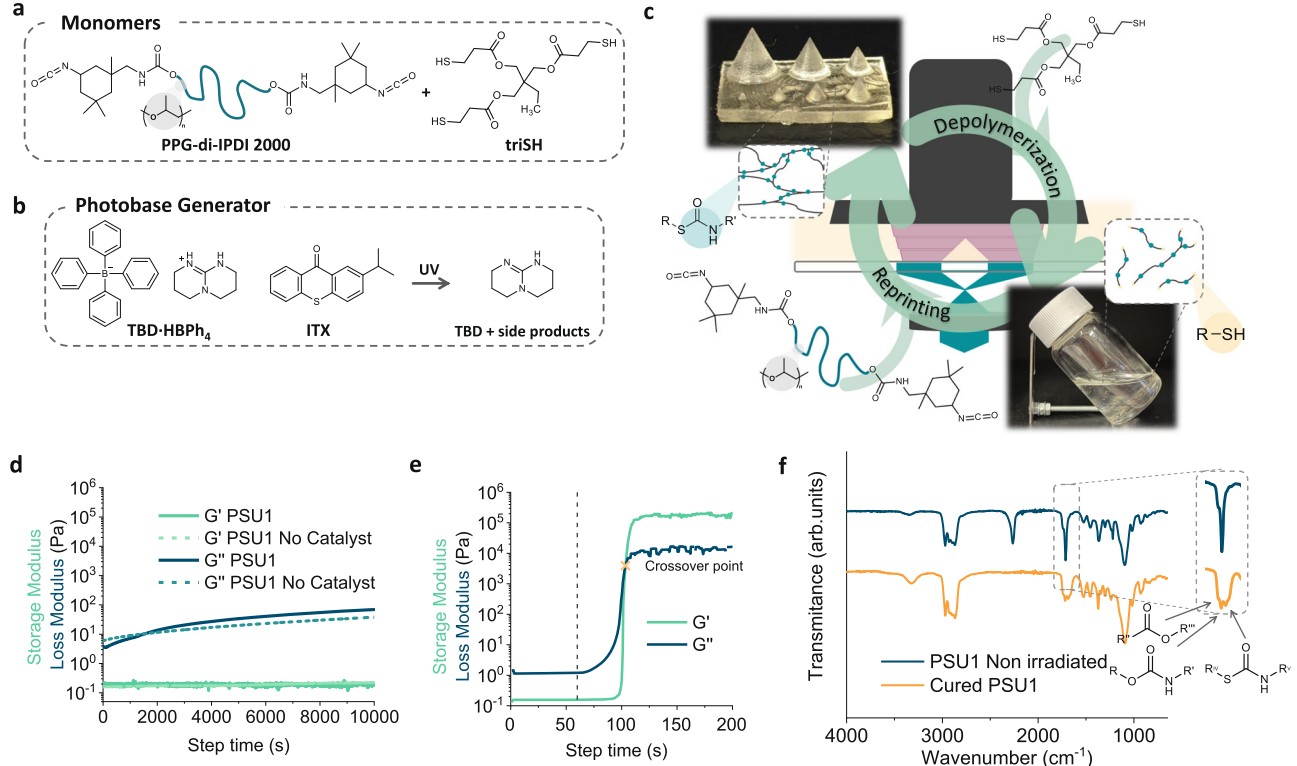

**Fig. 2 | Characterization of photocured poly(thio)urethane networks.**
**a** Chemical structures of the monomers and **b** PBG and sensitizer structure and photodecompositon reaction upon UV-light irradiation. **c** Reaction scheme of the depolymerization and loop-printing of PSU materials. **d** Evolution of the storage modulus (G′) (green) and the loss modulus (G″) (blue) over time of PSU1 with catalyst (continuous line) and without catalyst (dashed lines). **e** Evolution of G′ (green) and G″ (blue) of PSU1 resin as function of UV-LED (365 nm) irradiation time (started after 60 s experiment time) at an intensity of 20 mW/cm². Crossover point marked with a yellow cross. **f** IR spectra of PSU1 resin prior to (blue) and after irradiation (orange), most relevant bands are highlighted.

irradiance intensity to 200 mW/cm² (Supplementary Fig 1b) resulted in a substantially faster rise of the storage modulus related to the faster activation of the PBG, although the use of such high intensities will limit the application of the system in conventional 3D printers. Our reported system differs from previously reported PBG-catalyzed PSU polymerization, as fast photocuring profiles were obtained using UV-A medium irradiation in contrast to high-intensity Hg-Xe lamps, which are not suitable for the manufacturing technique[44].

To further demonstrate the versatility of the system we varied the molecular weight and nature of the isocyanate containing prepolymer. For that aim, IPDI terminated PPG ($M_w$: 1000 g/mol) (PPG-di-IPDI 1000), polytetrahydrofuran ($M_w$: 1000 g/mol) (PTHF-di-IPDI 1000) and polyethyleneglycol ($M_w$: 400 g/mol) (PEG-di-IPDI 400) were studied together with hexamethylenediisocyanate (HDI) terminated PPG ($M_w$: 1000 g/mol) (PPG-di-HDI 1000) (refer to "Methods" for synthetic details) to analyze the effect of the isocyanate source. All the tested formulations yielded thermosets upon UV light irradiation, although it was observed that the curing rate was substantially slower when decreasing the molecular weight of the prepolymer (Supplementary Fig. 2). We hypothesized that this observation was related to higher entanglement of higher molecular weight prepolymers inducing a faster rise of the storage modulus. By using an aliphatic primary PPG-di-HDI 1000 containing prepolymer, the crossover time was substantially reduced, which is related to the more reactive nature of the primary isocyanate of the HDI, however, a shorter pot life was observed. On the other hand, semicrystalline prepolymers such as PTHF1000 and PEG400 were also suitable for the preparation of thermoset PSU materials through this process, although cross-linking rates were substantially lower. Thermal properties of the prepared materials analyzed by differential scanning calorimetry (DSC) revealed glass transition temperatures ($T_g$) ranging from −50 °C to 2 °C which

showed the potential of acquiring different 3D objects using this approach (Supplementary Fig. 3). The versatility of the process was also shown by forming PSU networks bearing IPDI as the di-isocyanate, which yielded a printed object with higher cross-linking density and a $T_g$ of 58 °C (Supplementary Fig. 3).

Among the tested formulations PSU1 appeared to be the most suitable for implementation in the VP process due to its reasonably fast photocuring profile and good stability, and was therefore the formulation chosen for further investigation.

**Printing parameters and optimization**
Once the photocuring ability of PSU1 system was confirmed, its printability in a commercial DLP printer (Asiga Max-UV) emitting light at 385 nm was investigated. After calculating the inherent photonic parameters such as penetration depth ($D_p$) and critical exposure ($E_c$) of the resin PSU1 (Supplementary Fig. 4) according to Jacobs equation[48], we found that the ideal exposure times were 30 s of burn-in layer (first printed layer) and 20 s for the subsequent layers. The resolution of the printed objects was determined by printing custom designed specimens bearing engraved lines with widths varying between 1000, 500, 250, and 100 µm[49]. Then the samples were examined by optical microscopy (Zeiss AXIO Scope A1). All the features were clearly perceptible and retained the same shape, with ≈10% isotropic shrinkage after the development process related to solvent evaporation (Supplementary Fig. 5). To further confirm the resolution and the quality of the printed surfaces, cones with diameters between 5 and 1 mm were printed onto a rectangle and analyzed by scanning electron microscopy (SEM) (Fig. 3f). The ability of PSU1 to be printed without the addition of the volatile diluent was also investigated. With that aim, PSU1 resin was prepared as reported above and completely dried under vacuum prior to be printed. The prints resulted in 3D objects

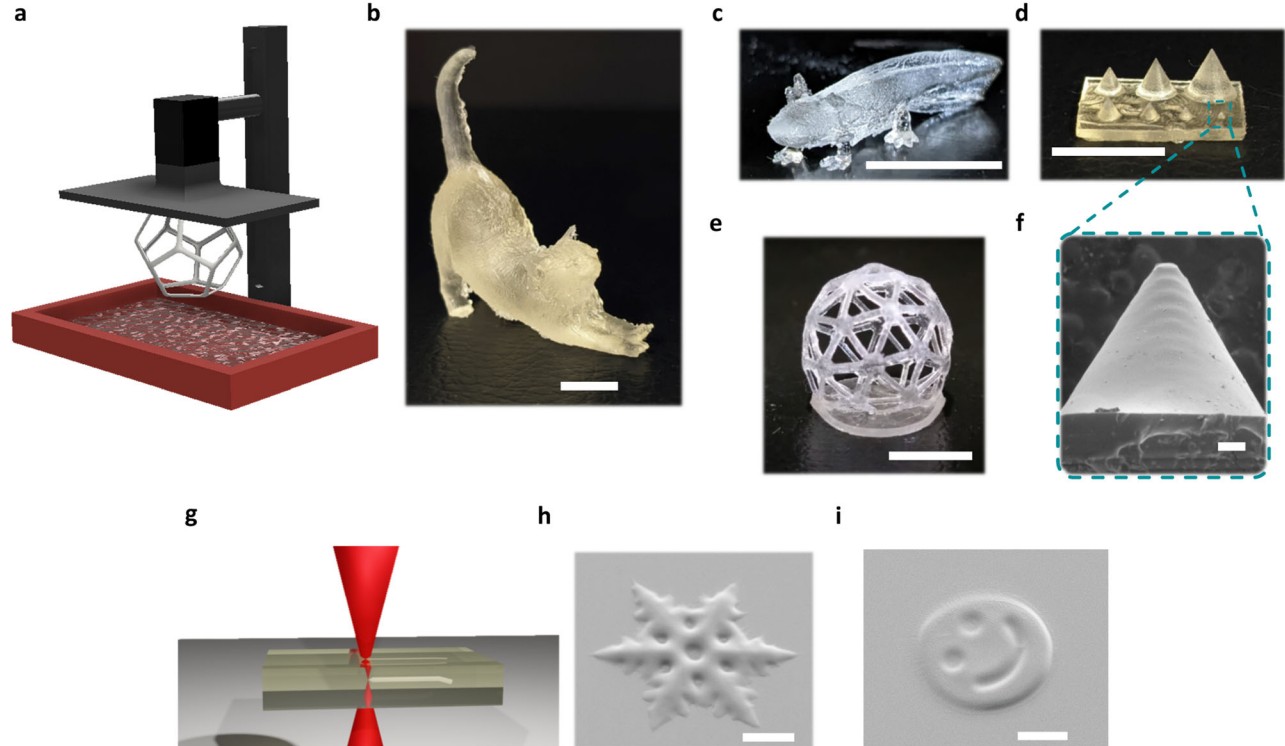

**Fig. 3 | 3D printing of PSU materials. a** Scheme of the DLP VP used for the printing of structures **b**–**e** Optical photographs of printed structures. Scale bar 1 mm. **f** SEM image of the smallest pyramid, scale bar 100 μm. **g** Two photon lithography DLW printing scheme. **h**, **i** SEM image of DLW printed samples, scale bar 30 μm.

with good resolution for low height objects (Supplementary Fig. 6). For higher objects (h > 3 mm) lower degree of resolution was achieved compared to for the analog of the resin containing the diluent. We attribute the loss of resolution to catalyst migration with leads to undesired polymerization on the surfaces. Once the successful printability of macroscopic structures was confirmed, microscale printing was targeted using two-photon direct laser writing (DLW) technique. It should be noted that the use of photobase generators as a polymerization phototrigger using a two-photon process has not been widely explored until now. With this method, we successfully laser printed microscale structures at high speeds (2 mm/s, 4–5 min total print time) and with excellent surface finish. In initial attempts, low resolution of printed structures was attributed to in-situ migration of the catalyst between the polymerized printed structure and the surrounding unpolymerized resin (Supplementary Fig. 7). To improve this, octanoic acid (10 wt% with respect to the PBG) was added to the resin as a catalyst quencher, and furthermore, the laser printed samples were developed immediately after printing in a solution of acetone containing excess benzoic acid. This process resulted in a noticeable improvement of the printing resolution. 3D microstructures were successfully printed using DLW (Fig. 3h, i).

### Repairability of PSU printed objects

As mentioned above, due to the cross-linked nature of traditional 3D printed photopolymers, they cannot be repaired if they are damaged during their use, which leads to a large amount of waste. By the incorporation of thermally activated DCBs in the printed structures, damaged objects could be repaired by healing them with heat. In this sense, thiourethane bonds have been reported to undergo dissociative bond exchange at mild temperatures and stoichiometric conditions, which has been used to reprocess poly(thio)urethane cross-links (Fig. 4e)[50–52]. To demonstrate the healing character of PSU1 material, we scratched a printed film and placed it between glass slides at 120 °C. The complete healing of the scratch was observed within 12 h with an

optical microscope (Fig. 4c). Stress relaxation studies revealed a significant effect of the temperature on the relaxation rate with an activation energy based on the Arrhenius equation for the thiourethane exchange of 90 kJ/mol which is consistent with previous works on similar structures[50,51]. Most importantly, the dissociative dynamic mechanism of the thiourethane network was confirmed with FTIR spectroscopic studies at 120 °C where an increase of the transmittance of the isocyanate stretching band over time was readily observed, and where a maximum of the band was achieved after 68 min heating (Supplementary Fig. 9). As proof of concept and to demonstrate the effective repairability of the reported resin, we 3D printed an axolotl that was significantly damaged on the tail (Fig. 4a). To fix this damaged object, first, the area of the broken part was polished to obtain a flat surface, then a newly printed tail (with Sudan I as a dye for clarity) was placed, applying gentle pressure with the hands. Then this piece was introduced in an oven at 120 °C for 1 h to enhance the dynamic bond interaction and heal the damaged object. To further analyze the repairability of the 3D printed objects, rectangular strips were printed bearing a hole in the middle (damage) (Fig. 4b). The holed sample was filled with new resin mixture and irradiated with UV light (365 nm) at an irradiance of 20 mW/cm² for 30 min. Afterward, the filled strip was placed between two glass covers held by metal binder clips and introduced in an oven at 120 °C for 1.5 h. For comparison, other set of repaired strips was not thermally cured. Then the mechanical properties of the control samples were compared to the repaired samples through uniaxial tensile testing. The repaired samples achieved 95% of the strain at break while 96% of the stress at break was recovered. However, then the repaired samples were not thermally treated, only 75% of strain at break and 76% of stress at break was observed. The robustness of the repairing process is shown by the fact that the fracture occurs across the repaired circle rather than at the edge of the repaired part. When testing the holed (damaged) sample, only 69% of the elongation at break was achieved while mechanical strength remained 49% of the control sample, indicating the effective

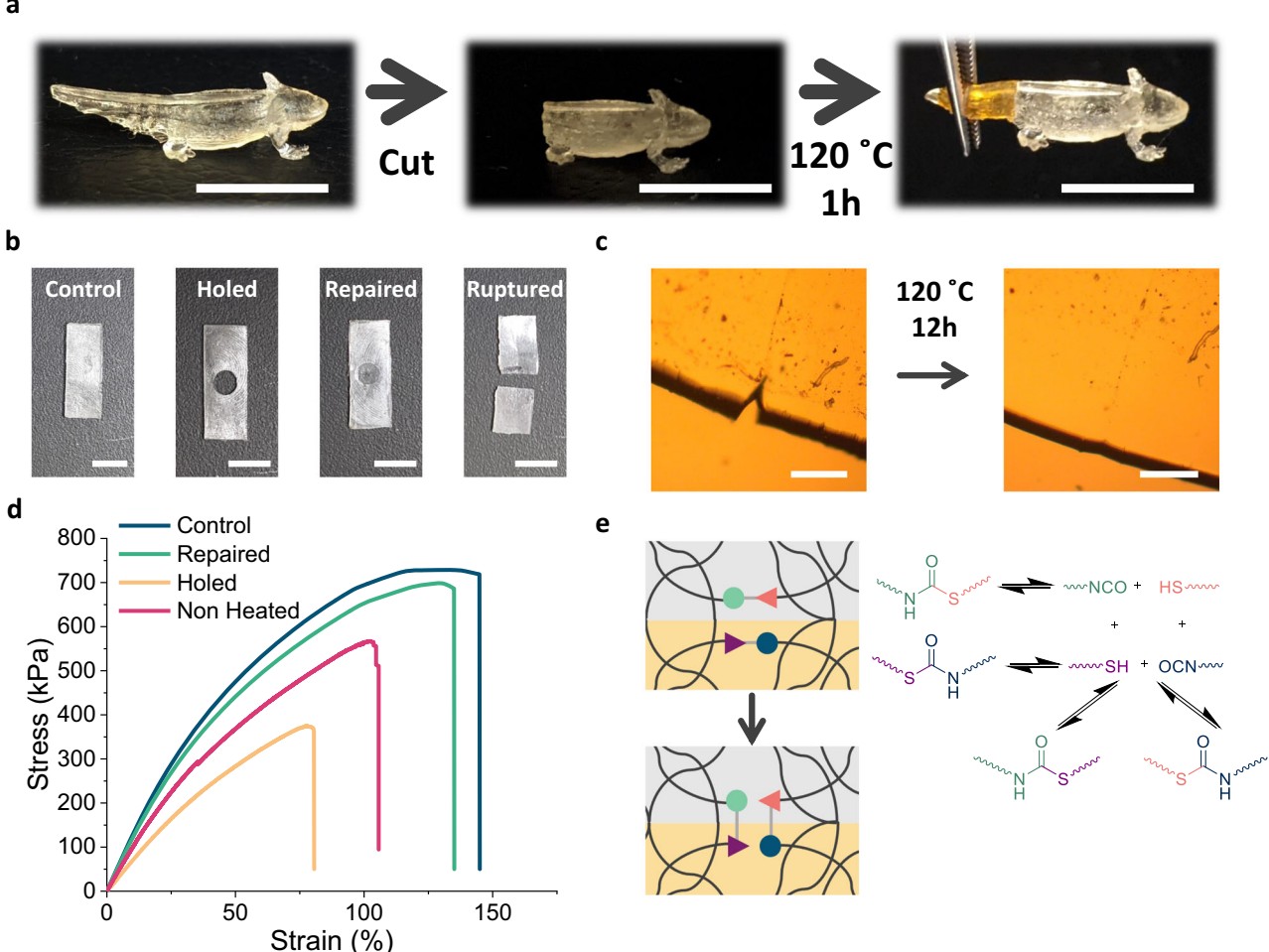

**Fig. 4 | Repairing PSU materials. a** Repair of the axolotl's tail by addition of self-healing tail. **b** Photographs of control sample, sample printed bearing a hole, repaired sample and repaired and fractured sample. Scale bar 7.5 mm **c** Self-healing experiment at 120 °C after 12 h. Scale bar 200 μm. **d** Uniaxial stress–strain curves for the control sample (blue), repaired sample (green), holed sample (orange) and Non cured sample (pink). **e** Healing scheme of thiourethane bonds through dissociative mechanism.

reparation of the damaged specimen, allowing recovery to almost the initial mechanical performance (Fig. 4d). These observations demonstrate the first use of thiourethanes as effective DCBs for healing and repair of printed objects.

**Closed-loop recyclability of 3D printed objects (decross-linking of thiourethane bonds via thiol-isocyanate exchange reaction)**
One of the biggest challenges to increase the sustainability of 3D printed thermoset materials is their limited recyclability. The strong covalent bonds generated during the printing process, generally through the free radical polymerization (FRP) of (meth)acrylate monomers, yield non-degradable C-C bonds in the polymer network. Although some examples showed the possibility to reuse 3D photoprinted materials into new products with retained material properties[53,54], to the best of our knowledge no work pointed to the recycling of the photoprinted objects into liquid resins that can be re-printed. Encouraged by this challenge and the recent findings where traditionally prepared PSU thermosets were depolymerized into thiol-terminated oligomers at room temperature[33,55], we pursued the chemical depolymerization of PSU1 3D printed structures into thiol terminated resins that could be 3D re-printed leading to a zero waste closed-loop photoprinting platform. To confirm this hypothesis, we cut 3D printed objects into small pieces that were mixed together with 2 equivalent excess of the triSH in acetone and 0.1 wt% of 1,1,3,3-tetramethylguanidine (TMG) as basic catalyst (refer to "Methods"). By stirring this mixture at room temperature, complete dissolution of the pieces was observed within 4 h. Depolymerization kinetics showed a linear dependance of the depolymerization product formation and time, suggesting surface depolymerization due to the absence of induction time (Supplementary Fig. 10). After a simple acid wash to eliminate any base catalyst presence, the recycled thiol oligomer was recovered in high yields (>95%). It should be mentioned that by the addition of 19.5 wt% of triSH in each depolymerization cycle PSU1 totally depolymerizes into oligomeric structures. The depolymerization reaction proceeds via the base catalyzed associative dynamic mechanism of thiourethane bonds in presence of excess of thiols[33]. Upon addition of at least one molar excess of thiol functional group towards isocyanate functionality the stoichiometry balance is shifted, and the network could be depolymerized based on Flory-Stockmayer equation for the critical gelation point[33]. The molecular structure of the depolymerized oligomers was analyzed by ¹H-NMR and FTIR spectroscopy confirming the expected formation of thiol terminated (thio)urethane oligomers (Fig. 5b). The molecular weight of the depolymerized oligomers doubled the initial PPG-di-IPDI 2000 oligomer's molecular weight as determined by size exclusion chromatography (SEC) (Supplementary Fig. 11) due to the equilibrium of species in the mixture, thus yielding resin formulations with increased viscosity compared to the original PSU1 which were still feasible for the

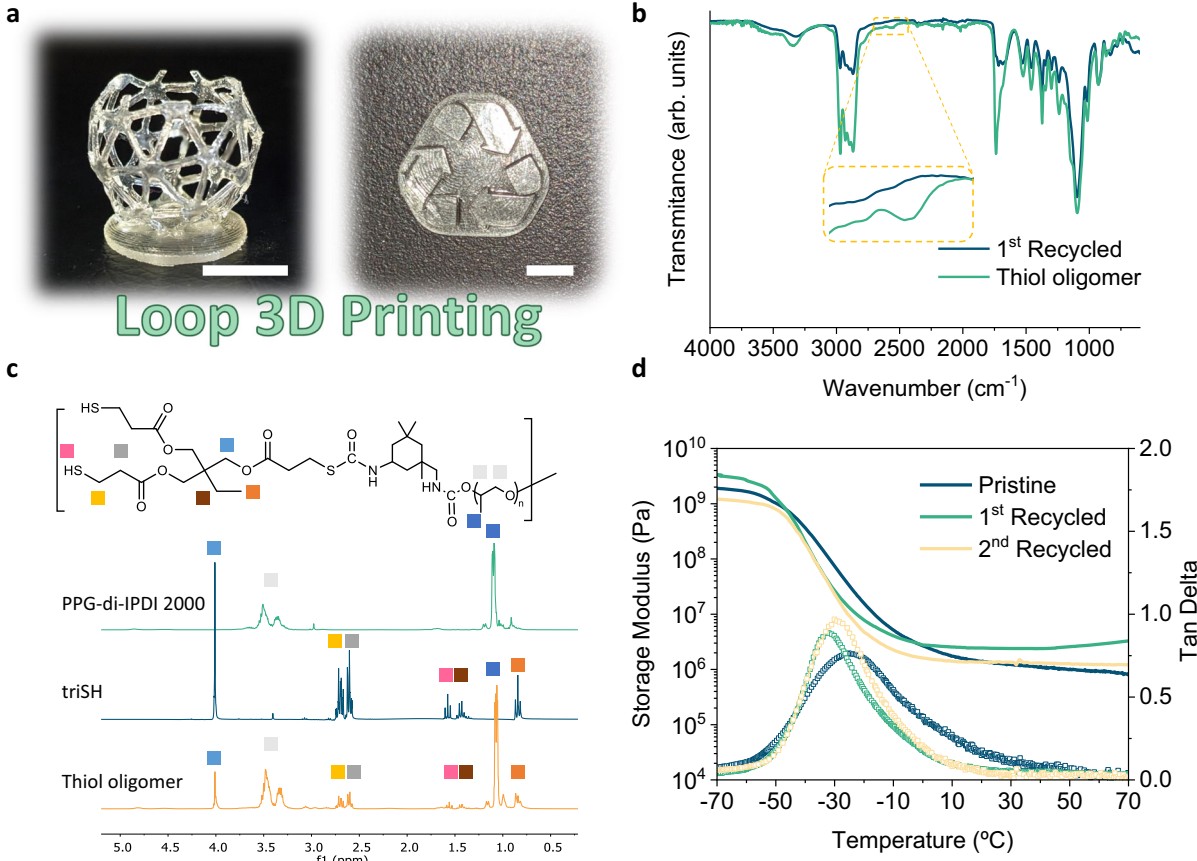

**Fig. 5 | Close-loop recycling of 3D printed polythiourethanes. a** Photographs of 3D printed structures using recycled resin. Scale bar 1 cm. **b** FTIR spectra of the 1st recycled PSU material (blue) and recycled thiol terminated resin (green). **c** $^1$H-NMR spectra of PPG-di-IPDI 2000 (green), triSH (blue) and recycled thiol terminated oligomer resin (orange). Proton assignment according to the idealized structure of the recycled thiol terminated oligomer. **d** Dynamic mechanical analysis of pristine (blue), 1st recycled (green) and 2nd recycled (orange) PSU thermosets. Storage modulus is shown as a line and tan delta using hollow squares.

printing process (Supplementary Fig. 20). It should be mentioned that PSU networks composed of IPDI and triSH were also susceptible to depolymerization under identical conditions although longer times were needed, resulting in oligomeric thiol terminated structures as shown by SEC (Supplementary Fig. 12). The depolymerization process was versatile for diverse compositions, thus enabling recyclability of printed objects of varied thermomechanical properties.

The addition of stoichiometric amount of PPG-di-IPDI 2000, 0.5 wt% TBD·HBPh$_4$ and 0.25 wt% ITX to the thiol oligomer after the quantification of the thiol content of the recycled oligomers by iodometric titration (Refer to "Methods"), and irradiation with UV light centered at 365 nm and an irradiance of 20 mW/cm$^2$ for 30 min, yielded the recycled materials which were characterized by FTIR spectroscopy (Fig. 5b). Two cycles of this process resulted in materials with similar thermomechanical properties in terms of glass transition temperature (T$_g$) and storage modulus between original and recycled PSU thermosets as shown in Fig. 5d. Moreover, thermogravimetric analysis (TGA) and gel content (GC) experiments further confirmed the cross-linked nature of the recycled polymer network (Supplementary Fig. 14 and Supplementary Table 2). Interestingly, we found that in order to obtain recycled PSU materials with retained mechanical properties it is of major importance to maintain the stoichiometric balance between the thiol and isocyanate groups (due to the step-growth nature of the polymerization) and therefore quantify the thiol presence on the depolymerized oligomers. In this regard, uniaxial tensile testing of printed dumbbell-shaped samples

showed mechanical property retainment, supporting the fact that chemical depolymerization allowed for the better property achievement after recycling cycles (Supplementary Fig. 17). The printing of PSU1 without acetone yielded slightly lower mechanical performance compared to the PSU1 material printed with acetone (Supplementary Fig. 18). This result was presumably attributed to lower monomer conversion due to gelation of the network and therefore this aspect would require further optimization and post-curing steps. Additionally, the photorheological behavior of the recycled resin was comparable to the photorheological profile of the pristine PSU1 resin (Supplementary Fig. 15) suggesting that similar printing conditions would be applicable. As shown in Fig. 5a, 3D structures were printed using the same printing conditions as the ones for PSU1. Hence, the reported poly(thio)urethane VP process enables the recovery of recycled thiol terminated oligomers by the depolymerization of printed objects with the addition of the starting thiol monomer, which can be closed-loop printed into materials allowing for a circular vat photoprinting process.

In summary, we have demonstrated a simple and versatile platform for recyclable and 3D printable resins based on a thiol-isocyanate polymerization reaction requiring minimal synthetic effort. In particular, the photobase generator activated formulation allowed the fabrication of both macro-featured and micro-featured structures in reasonable time via commercially available DLP or two photon DLW 3D printers, respectively. The stability and photorheological behavior along with the chemical and thermomechanical properties of the

prepared materials were studied. Additionally, the dynamic nature of the thiourethane bonds in the network backbone permitted the reparation and chemical recycling of printed structures with identical properties. In particular, the addition of a thiol monomer to printed samples in basic conditions generated thiol terminated oligomers which were utilized to produce reclaimed materials. This process allowed for the unprecedented closed-loop 3D printing of the resin formulations with reduced waste as a circular approach. We therefore believe that this contribution is a significant advancement in the field of sustainable AM.

## Methods
### Materials
Isophorone diisocyanate (IPDI) (>99%), isopropylthioxanthone (ITX) (>98%), Sudan-1 (>93%), and 1,8-diazabicyclo[5.4.0]undec-7-ene (DBU) (>98%) were purchased from Tokyo Chemical Industry (TCI) (https://www.tcichemicals.com/ES/en/). Hexamethylene diisocyanate (HDI) (≥98.0%), polyethylene glycol (PEG) ($M_n$: 400 g/mol), polypropylene glycol (PPG) ($M_n$: 2000 g/mol) and, ($M_n$: 1000 g/mol) 1,5,7-triazabicyclo[4,4,0]dec-5-ene (TBD), trimethylpropane tris(3-mercaptopropionate) (triSH) (≥95.0%), and 1,1,3,3-tetramethylguanidine (TMG) (99.0%), were obtained from Sigma-Aldrich (https://www.sigmaaldrich.com/ES/es). Sodium tetraphenylborate (NaBPh₄) (99%) was purchased from ABCR (https://abcr.com/de_en/). PPG-di-IPDI 2000 was kindly supplied by Cidetec (https://www.cidetec.es). TBD·HBPh₄ salt preparation: 1,5,7-Triazabicyclo[4.4.0]dec-5-ene (1 eq.) was dissolved in 10% HCl aqueous solution and dropwise added to a sodium tetraphenylborate (1.1 eq.) aqueous solution. The subsequent mixture was filtered and washed thoroughly with water and chilled MeOH. The photocatalyst was obtained as a white powder in quantitative amounts, and characterized by [1]H NMR. Isocyanate-terminated prepolymers were prepared by the addition of 2.1 eq. of isocyanate into 1 eq. of polyol previously dried under reduced pressure. The conversion of the reaction monitored by FTIR spectroscopy.

### Characterization techniques
Proton nuclear magnetic resonance ([1]H-NMR) spectroscopy data was collected on a 300 MHz Bruker Advance DPX Spectrometer at 20 °C. The samples were dissolved in deuterated solvents and chemical shifts reported in ppms and referenced to solvents peaks.

Fourier Transform Infrared Spectroscopy (FT-IR) spectra were recorded on a Nicolet iS20 Spectrometer using Attenuated Total Reflection (ATR) at a resolution of 2 cm⁻¹ and a total of 32 interferograms. The spectra at high temperature were obtained on a Nicolet 6700FT-IR spectrophotometer equipped with a specap variable temperature transmission cell. Spectra were recorded in the range of 4000 and 400 cm⁻¹ with a spectrum resolution of 4 cm⁻¹, and a total of 64 interferograms. Samples were prepared by photocuring of PSU1 on KBr windows as described and subsequently dried.

Thermogravimetric analysis (TGA) was carried out on a Q500 Thermogravimetry Analyzer from TA instruments. The system was heated with a heating rate of 10 °C/min from room temperature to 600 °C under atmospheric conditions.

Differential Scanning Calorimetry (DSC) analyses were performed on a DSC Q2000 from TA Instruments. The analysis was performed in sealed aluminum pans, under dry $N_2$ atmosphere using a heating rate of 10 °C/min a heating ramp from −80 to 160 °C.

Size exclusion chromatography (SEC) measurements were carried out on a Waters 1515 GPC chromatograph using chloroform as eluent. Molar masses are referred to PS standards.

Gel content of the samples was determined gravimetrically. At least two specimens of each sample were Soxhlet extracted for 24 h in refluxing THF. The insoluble fraction was dried in the oven. The gel content is expressed as the percentage of insoluble fraction in the sample.

Dynamical Mechanical Thermal Analysis (DMTA) were recorded on a Triton Tritec 2000 DMA instrument. Specimens with dimensions of 15 mm × 5 mm × 0.5 mm were tested in bending mode at a amplitude of 50 μm and a frequency of 1 Hz. A heating ramp of 4 °C/min from was used from −70 to 70 °C. The glass transition temperatures ($T_g$) were calculated from the peak of tan δ.

Rheology measurements were conducted on a AR-G2 rheometer from TA Instruments. For photorheology experiments, a LED UV-curing accessory centered at 365 nm was used together with an acrylic transparent bottom plate. Photorheology experiments were performed at a frequency of 1 Hz and in the linear viscoelasticity region using 20 mm parallel plates and a gap of 400 μm. To determine the viscosity of the formulated resins, a 1° 40 mm cone geometry was used. The shear rate window was from 1 to 100 s⁻¹ collecting 11 data points during the experiment. Average values of at least three measurements are shown.

Stress relaxation measurements were performed in a ARES rheometer (Rheometrics) at different temperatures using a film tension fixture and 5% strain. The samples used for this measurement were in the range of 2.5 to 3 mm wide and 0.5 to 1 mm thick.

Mechanical properties for repair experiments were determined on a Instron 5569 instrument including a 100 N cell at room temperature by uniaxial tensile test. Printed or dye cut samples (20 mm × 7.5 mm × 1 mm) were tested at 2 mm/min strain rate, being the distance between the grips of 10 mm. The Young's modulus (E) was calculated from the initial slope of the stress–strain curve and the deformation at break from the maximum value of stress.

Mechanical property characterization of pristine and recycled resins was performed on a TA.HD Plus Texture Analyzer from Texture Technologies, bearing a load cell of 30N at room temperature by uniaxial tensile test. Printed or casted dumbbell-shaped samples, according to ISO37 type IV specifications were used. The samples were attached to vice grips and subjected to increasing strain at a rate of 0.09 mm/s until mechanical failure.

The prepared resins were printed into 3D objects on a commercial Asiga Max-UV DLP 3D printer with a LED source centered at 385 nm. Computer designed 3D objects were prepared for printing in the Asiga Composer software. The optimized printing parameters were 30 s burn-in layer, 20 s exposure time and 50 μm layer thickness at an intensity of 20 mW/cm². The material without acetone was printed using 2 payers of 60 s burn-in and 15 s exposure time and 50 μm layer thickness at an intensity of 20 mW/cm². The printed objects were washed thoroughly with acetone to remove unreacted monomers and postcured in an oven at 70 °C overnight.

Scanning electron microscopy (SEM) measurements were conducted on a Hitachi TM3030Plus microscope at an accelerating voltage of 15 kV. The printed samples were mounted on a specimen stub attached with double-sided carbon adhesive tape and coated with gold before being observed. For samples printed using DLW SEM measurements were performed with Zeiss Supra 55VP (Carl Zeiss AG) at 3 kV in secondary electron mode. Prior to imaging, the structures were sputter-coated with a 12 nm layer of Pd:Pt.

Two-photon 3D laser printing (2PLP) was performed using a commercial Photonic Professional GT 2 (Nanoscribe GmbH) direct laser writing system. The fabrication of all microstructures was carried out in oil immersion mode using a 63x oil immersion objective (NA = 1.4). GWL files for desired geometries were generated from STL files of 3D structures employing commercial Describe software (Nanoscribe). Slicing was set to 0.3 μm and hatching to 0.2 μm. Laserpower was 50 mW or 40 mW for the snowflake and smiley respectively, and the scan speed was 2000 μm/s. The laser printed samples were developed in a solution of acetone with excess benzoic acid immediately after printing, then washed with acetone, followed by drying under ambient conditions. The printed samples were stored under ambient conditions. The entire process was performed under yellow light conditions.

## Sample preparation

In a typical photopolymerization, equimolar amounts of isocyanate and thiol cross-linker were mixed together with TBD·HPBh$_4$ (0.5 wt%) and ITX (0.25 wt%) solution in acetone (30 wt%). The mixture was vortex mixed until homogeneous colorless resin was obtained. This resin was poured into a silicone cast, covered with glass slides and irradiated with a UV-light centered at 365 nm with 60 mW/cm$^2$ light intensity for 15 min. Afterward, the samples were postcured at 70 °C overnight.

As an example, for the preparation of PSU1, 1000.00 mg of PPG-di-IPDI 2000 and 109.00 mg of trimethylolpropane tris(3-mercapto-propionate) triSH were mixed in vial. On the other hand, 5.50 mg of TBD·HBPh$_4$ photocatalyst and 2.75 mg of sensitizer (ITX) were dissolved in 300 mg of acetone and added to the monomer mixture. This mixture was vortex agitated until homogeneity was achieved. This resin was freshly used for subsequent studies.

3D printed structures were washed with acetone to remove unreacted monomer prior the postcuring in an oven at 70 °C overnight.

Substrate functionalization for 2PLP: To improve adhesion of 3D printed microstructures, glass slides (Marienfeld, 170 ± 5 μm strength) were washed first with isopropanol, then with acetone and dried using pressurized N$_2$. Subsequently, the surface was cleaned and activated for 1 min by plasma treatment. Following this, the glass slides were immersed in a $4 \times 10^{-3}$ M solution of 3-(trimethoxysilyl) propyl acrylate in toluene for 1.5 h. After washing twice in toluene and once in acetone and successive drying with pressurized N$_2$, the acrylate-functionalized glass slides were used as substrates for 2PLP microfabrication. The entire substrate functionalization was performed under yellow light conditions, along with subsequent storage.

Sample preparation for 2PLP: Using a commercial 2PLP sample holder (Nanoscribe) for oil immersion mode acrylate-functionalized glass slides were attached by tape. Immersion oil was added on the unfunctionalized glass slide surface. A droplet of the functional ink was placed in the center of the slide. The sample was utilized directly for microprinting.

The resin formulation was prepared for 2PLP similarly to the PSU1 with the addition of octanoic acid ($3 \times 10^{-4}$ mmol, 10% w.r.t. the PBG was added to the formulation. The resin mixture was shaken until homogeneity was achieved. The acetone was removed under reduced pressure and the resin was used for printing.

## Healing test

Printed samples of 10 mm × 10 mm × 0.1 mm dimensions were scratched using a blazed blade, placed between two glass slides, and clamped using metal binder clips. The samples were placed in a temperature-controlled microscope stage (Linkam LTS420E) at 120 °C temperature while images were taken by an optical microscope (Zeiss AXIO Scope A1) every 30 min.

## Repairability

Control and tested samples were 3D printed using the PSU1 formulation and optimized printing conditions. Specimens were 20 mm × 7.5 mm × 1 mm and the tested sample was printed with a 4.5 mm diameter hole in the middle. The repair was carried out by filling the hole with the PSU1 resin and irradiated with a UV-light centered at 365 nm with 60 mW/cm$^2$ light intensity for 15 min. The samples were postcured at 70 °C for 1 h and at 115 °C for 1 h while applying gentle pressure with binder clips.

## Depolymerization of PSU1

For the depolymerization of the cross-linked poly(thio)urethane material, 2.00 g of PSU1 (1.47 mmol thiourethane [S-CO-NH]) was cut into small pieces and mixed with 2 molar equivalent excess of thiol groups from trimethylolpropane tris(3-mercaptopropionate) (0.39 g, 2.95 mmol thiol [SH]), 0.1 wt% of TMG ($2.39 \times 10^{-3}$ g, 0.02 mmol) and 20 mL of acetone. The mixture was stirred for 3 h at room temperature until the film was completely dissolved. Solvent was evaporated and the residue redissolved in 20 mL of DCM. This solution was extracted with 1 M HCl solution and with brine solution. The organic phase was dried over anhydrous Na$_2$SO$_4$ and solvent evaporated under reduced pressure to yield the recycled thiol terminated oligomer in quantitative yield. For conversion calculation, 25 mol% of toluene (according to thiourethane bonds) was added as an internal standard. Depolymerization conversion was calculated towards the appearance of PPG derived signals.

## Quantification of thiols by iodometric titration

To quantify the presence of thiol functionalities on the recycled oligomers, an adapted procedure from literature was used[56]. To an Erlenmeyer flask containing 5 mL of DCM, 0.5 mL of glacial acetic acid, and 20 mL of 0.1 M KI(aq), ~50.0 mg of I$_2$(s) were added and stirred to ensure complete dissolution of the iodine. To this mixture, ~100.0 mg of sample (1$^{st}$ Recycled Oligomer or 2$^{nd}$ Recycled Oligomer) dissolved in 5 mL of DCM were added and vigorously stirred over 5 min. To the mixture, 5 drops of aqueous starch indicator (1%) was added and titrated using 9.0 mM Na$_2$S$_2$O$_3$ aqueous solution while vigorously stirring during the titration until complete discoloration. The titration was performed three times for each sample. The amount of thiol groups in the formulation was calculated according to the following equation:

$$SH\ moles/g = \frac{(2xI_2\ mol) - N_2S_2O_3}{g\ of\ sample} \qquad (1)$$

## Repolymerization

The recycled oligomers (2.95 mmol thiol [SH]) were recross-linked by adding stoichiometric amounts of PPG-diIPDI (M$_n$: 2444.6 g/mol) (2.95 mmol isocyanate [NCO]), 0.5 wt% of TBD·HBPh$_4$ and 0.25 wt% ITX dissolved in 30 wt% of acetone according to the complete system similarly to the original formulation. The mixture was stirred until homogeneity was achieved. This resin was freshly used for subsequent studies.

## Data availability

Data will be available upon request.

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

## Acknowledgements

X.L.P. gratefully acknowledges the support from the Spanish Ministry of Universities (FPU18/04904). The authors thank C. Vazquez-Martel and B. Weidinger (Heidelberg University) for their support in SEM imaging, N. Ramos-Gomez for the support on healing experiments, SGIker (UPV/EHU/ERDF, EU) for technical and human support, and all the Research Staff from the Rheology Group and Polymer Processing Group at POLYMAT (UPV/EHU) for their support regarding rheological and thermal-mechanical characterization.

## Author contributions

X.L.P. and H.S. conceived and initiated the study. X.L.P. and O.V. synthesized the materials and carried out the DLP printing, repairing, and recycling experiments. S.O.C. performed the DLW experiments and SEM analysis. E.B., T.E.L., and H.S. motivated and supervised the research program. X.L.P., T.E.L., and H.S. wrote a first draft of the manuscript. All authors discussed the results and worked on the manuscript.

## Competing interests

The authors declare no competing interests.
