## [Peer Review File · Nature Communications]

Recyclable photoresins for light-mediated additive manufacturing towards Loop 3D printingEditorial Note: This manuscript has been previously reviewed at another journal that is not operating a transparent peer review scheme. This document only contains reviewer comments and rebuttal letters for versions considered at Nature Communications.

Reviewers' Comments:

Reviewer #1:

Remarks to the Author:

Please see uploaded file for my comments on how the authors have addressed my original comments. I was also asked to assess the responses to reviewer #2's comments. Having studied the comments and responses, all of these comments appear to have been addressed thoroughly and satisfactorily.

Most of my original comments have been satisfactorily addressed. However, I still have the following minor concerns:

My Original Comment: Reporting a linear speed for the scanning beam of the two photon printing may not be the best way to help the user understand throughput. It may be better to report a volumetric material processing rate instead or as well.

Author Response: Linear scan speed in μm or mm s^{-1} (as well as laser power in mW) is the standard parameter given for two-photon laser printing since it is the direct input required for the use of the printers. This is well established in the community. The total printing time, as calculated by the printing software, was 4 min 40 s and 5 min for the smiley and snowflake, respectively. For clarity and to give a better idea about the speed of the process, we have now added total print time to the manuscript. The modified text can be found in the section 'Printing parameters and optimization'.

"With this method, we successfully laser printed microscale structures at high speeds (2 mm/s, 4-5 min total print time) and with smooth surfaces.

My further comment: the reader could be further helped by stating the volume of the structures that could be printed within 4-5 minutes.

My original comment: The description of the addition of octanoic acid as a quencher plus rapid development is said to result in a "significant" improvement in the printing resolution, yet no numerical values for resolution (e.g. minimum printable feature size) are shown, nor therefore is any statistical analysis that could justify a claim of significant improvement. If such data are available, I suggest they are added to the SI; otherwise, I suggest the word "significant" is changed.

Author Response: The noticeable printing resolution improvement by the addition of octanoic acid is supported by the Figure S7 added in the SI. There, the lower resolution of the printed constructs is clearly observed compared with the resolution of the samples using octanoic acid as quencher. We have modified the text in section 'Printing parameters and optimization' and removed the word significant.

In initial attempts, low resolution of printed structures was attributed to in situ migration of the catalyst between the polymerized printed structure and the surrounding unpolymerized resin (Figure S7). "To improve this, octanoic acid (10 wt% with respect to the PBG) was added to the resin as a quencher, and further the laser printed samples were developed immediately after printing in a solution of acetone containing excess benzoic acid. This resulted in a noticeable improvement of the printing resolution."

My further comment: Thank you for the additional explanation, which helps. Fig S7 is not totally convincing, as they are optical micrographs (which also look somewhat out-of-focus to me) whereas they are offered as a comparison to SEMs of the structures printed with octanoic acid in Fig 2h-i of the main text. So it is not surprising the structures made with the octanoic acid look sharper! If possible I would recommend putting SEMs in Fig S7, and showing the structures printed with/without octanoic acid side by side and with the same scale and imaging method.

Reviewers' Comments:

Reviewer #1:

Remarks to the Author:

Please see uploaded file for my comments on how the authors have addressed my original comments. I was also asked to assess the responses to reviewer #2's comments. Having studied the comments and responses, all of these comments appear to have been addressed thoroughly and satisfactorily.

Reviewer #3:

Remarks to the Author:

The authors have made the following changes in the manuscript's main text.

- Added a comment on how the current system is novel compared to previous work
- Added printed objects without solvent
- Added tensile test data for two cycles of recycled materials and compared them with pristine materials

However, the manuscript still does not provide convincing scientific evidence supporting the novelty and impact of the presented work. Furthermore, this work builds upon the previous work of Bowman et al. (Ref 29), which bears the questions of sufficient novelty to meet the standard expected for Nature Communications. On that basis, I recommend that this work be published in a more polymer-specialized journal, such as Polymer Chemistry or Macromolecules.

In terms of novelty, combining three old concepts doesn't make it a new idea. As pointed out by other reviewers as well, the originality of this work needs to be addressed. In the introduction, the authors addressed this concern by having one sentence pointing out, "In comparison...into 3D structures.". Firstly, the authors should reference the "other works with dynamic behavior" for comparison. Secondly, this argument is not valid. Wang et al. (Recyclable thermosetting polymers for digital light processing 3D printing. *Materials & Design*, 197, 109189.) and Zhao et al. (Reprintable polymers for digital light processing 3D printing. *Advanced Functional Materials*, 31(9), 2007173.) both show the ability of chemically recycling of the printed parts and re-printability the recycled resin. Although, the authors offer a different chemistry to achieve a similar recycling behavior. However, dynamic thiourethane chemistry has been well studied. (ref 29 and ref 47). Thirdly, the authors argue that the "photo base has to be designed for the process" in the response letter to emphasize the novelty. However, the photo base was synthesized based on the previous reports (Ref 41 and Ref 42). Moreover, the kinetic of the thiol-isocyanate reaction catalyzed by photo base has been well studied by Ley et al. (Ref 40). Overall, the manuscript reads like a report, and everything is as expected. There is no new knowledge or new aspect of fundamental understanding that the readers can get from the current report.

In addition, there are some major concerns I have for the manuscript:

1. The authors specifically choose a very loosely crosslinked system. The glass transition is around -30OC. How would the crosslinking density impact the materials in the depolymerization process? What is the mechanism of depolymerization process? Bulk or surface degradation? And what is the impact of the solvent (in this case, acetone) on the depolymerization kinetics? Would other solvent work for the depolymerization process?
2. Fig 3e, for the healing mechanism, the authors should comment on why it is a dissociative mechanism versus an associative mechanism. An FTIR or NMR spectrum showing intermediate thiol and isocyanate peaks at elevated temperatures should be provided to support this argument.

3. The authors show the printability of the resin without solvent. However, the resolution in Fig S6 looks far below the "good resolution" standard for DLP and two-photon lithography. The authors should also comment on the impact of the solvent on the mechanical properties. The tensile properties printed with and without acetone should be provided.

4. The resins contain free isocyanate, a highly toxic and unstable species. Therefore, the authors should comment on the limitation of the resins.

5. For Fig S13, the toughness of the material reduced dramatically with the recycled materials. For example, the 2nd recycled material shows ~ 50% reduction in the elongation/toughness. One unique advantage of the chemically recycling strategy is the ability to achieve similar mechanical properties. However, the results in the tensile data failed to provide the evidence for retaining the similar mechanical behavior.

Referee #1

Comment: Most of my original comments have been satisfactorily addressed. However, I still have the following minor concerns:

Response: We thank the reviewer for taking into account our new experiment to satisfy the concerns of the reviewer.

My Original Comment: Reporting a linear speed for the scanning beam of the two photon printing may not be the best way to help the user understand throughput. It may be better to report a volumetric material processing rate instead or as well.

Response: Linear scan speed in μm or mm s^{-1} (as well as laser power in mW) is the standard parameter given for two-photon laser printing since it is the direct input required for the use of the printers. This is well established in the community. The total printing time, as calculated by the printing software, was 4 min 40 s and 5 min for the smiley and snowflake, respectively. For clarity and to give a better idea about the speed of the process, we have now added total print time to the manuscript. The modified text can be found in the section 'Printing parameters and optimization'.

"With this method, we successfully laser printed microscale structures at high speeds (2 mm/s, 4-5 min total print time) and with smooth surfaces".

Comment: the reader could be further helped by stating the volume of the structures that could be printed within 4-5 minutes.

Response: We thank the reviewer for their comments. First, we would like to highlight DLW is a suitable technique for 3D printing fine featured structures at the microscale, when high resolution and precision are required, rather than high throughput output that may be targeted for other volumetric type 3D printing methods. Thus, since the volume printed is very small and depends of several parameters (see below), it is not common to given numbers in terms of volume per time, but rather scan speed and laser power are the usual reported parameters as indicated in the previous reply. To answer the reviewer's question, we have calculated from the STL file the volume of the Smiley structure (80 μm diameter) to be 28 530 μm^3 . Thus, using the reported parameters we printed at approximately 6500 μm^3 per min with these parameters. It should be noted, that depending on the ink used, the volume of a resultant structure can also vary from the initial print job, as each material can undergo shrinkage during development. Also, the print time for a single structure varies greatly as it is dependent on many factors, for example the directionality/offset of the print (i.e. each layer printed parallel or perpendicular to one another), the hatching and slicing distances of the structure, whether a core-shell method is employed, the magnification that is employed etc... Thus, we respectfully believe that for this specific technique reporting the volume that can be printed in an specific time would not benefit the reader.

My original comment: The description of the addition of octanoic acid as a quencher plus rapid development is said to result in a "significant" improvement in the printing resolution, yet no numerical values for resolution (e.g. minimum printable feature size) are shown, nor therefore is any statistical analysis that could justify a claim of significant improvement. If such data are available, I suggest they are added to the SI; otherwise, I suggest the word "significant" is changed.

Response: The noticeable printing resolution improvement by the addition of octanoic acid is supported by the Figure S7 added in the SI. There, the lower resolution of the printed constructs is clearly observed compared with the resolution of the samples using octanoic acid as quencher. We have modified the text in section 'Printing parameters and optimization' and removed the word significant and now reads as follows:

"In initial attempts, low resolution of printed structures was attributed to in situ migration of the catalyst between the polymerized printed structure and the surrounding unpolymerized resin (Figure S7). To improve this, octanoic acid (10 wt% with respect to the PBG) was added to the resin as a catalyst quencher, and furthermore, the laser printed samples were

developed immediately after printing in a solution of acetone containing excess benzoic acid. This process resulted in a noticeable improvement of the printing resolution.”

My further comment: Thank you for the additional explanation, which helps. Fig S7 is not totally convincing, as they are optical micrographs (which also look somewhat out-of-focus to me) whereas they are offered as a comparison to SEMs of the structures printed with octanoic acid in Fig 2h-i of the main text. So it is not surprising the structures made with the octanoic acid look sharper! If possible I would recommend putting SEMs in Fig S7, and showing the structures printed with/without octanoic acid side by side and with the same scale and imaging method.

Response: We thank the reviewer for this suggestion and have modified the Supporting Information Figure S7 to rather include an SEM image of the structure with and without octanoic acid to show the improved printing resolution. To further illustrate this point, the initial STL file used for printing has a diameter of 80 μm . With octanoic acid we measure a diameter of the structure of approximately 76 μm while the sample that does not contain octanoic acid has a diameter of approximately 94 μm .

Now Figure S7 Looks the following:

Figure S7: SEM images of DLW printed smiley without octanoic acid (left) and using octanoic acid (right). Scale bar 30 μm . Original STL file has a diameter of 80 μm . The sample printed without octanoic acid has a diameter of $\approx 96 \mu\text{m}$ while the sample printed with octanoic acid has a diameter of $\approx 76 \mu\text{m}$.

Reviewer #3

Comment: The authors have made the following changes in the manuscript's main text.

- Added a comment on how the current system is novel compared to previous work
- Added printed objects without solvent
- Added tensile test data for two cycles of recycled materials and compared them with pristine materials

Response: We thank the reviewer because it took us some time to optimize the conditions to make the required experiments and he valued the work done.

Comment: However, the manuscript still does not provide convincing scientific evidence supporting the novelty and impact of the presented work. Furthermore, this work builds upon the previous work of Bowman et al. (Ref 29), which bears the questions of sufficient novelty to meet the standard expected for Nature Communications. On that basis, I recommend that this work be published in a more polymer-specialized journal, such as Polymer Chemistry or Macromolecules. In terms of novelty, combining three old concepts doesn't make it a new idea. As pointed out by other reviewers as well, the

originality of this work needs to be addressed. In the introduction, the authors addressed this concern by having one sentence pointing out, "In comparison...into 3D structures.". Firstly, the authors should reference the "other works with dynamic behavior" for comparison. Secondly, this argument is not valid. Wang et al. (Recyclable thermosetting polymers for digital light processing 3D printing. *Materials & Design*, 197, 109189.) and Zhao et al. (Reprintable polymers for digital light processing 3D printing. *Advanced Functional Materials*, 31(9), 2007173.) both show the ability of chemically recycling of the printed parts and re-printability the recycled resin. Although, the authors offer a different chemistry to achieve a similar recycling behavior. However, dynamic thiourethane chemistry has been well studied. (ref 29 and ref 47). Thirdly, the authors argue that the "photo base has to be designed for the process" in the response letter to emphasize the novelty. However, the photo base was synthesized based on the previous reports (Ref 41 and Ref 42). The works covering the use of dynamic-covalent bonds on 3D printing structures Moreover, the kinetic of the thiol-isocyanate reaction catalyzed by photo base has been well studied by Ley et al. (Ref 40). Overall, the manuscript reads like a report, and everything is as expected. There is no new knowledge or new aspect of fundamental understanding that the readers can get from the current report.

Response: We thank the reviewer for his comments in regard to our manuscript. However, we respectfully believe that the novelty and impact of the present work would meet the requirements of Nature Communications.

Although our earlier research and others have reported the versatility of poly(thiourethanes) together with organic base catalysis (Ref 44), the design of a composition that is amenable to efficient photo-chemical processes for vat photopolymerization remains unprecedented. Moreover, fine tuning and fundamental understanding of the suitable rheological behavior coupled with photo-kinetics was paramount to enable the printing of fine resolution objects. The ability to prepare compositions over a broad range of glass transition temperature was also shown, and the repair of the printed object as well as efficient chemical depolymerization was shown for the printed objects.

To the novelty of the work has been highlighted by discussing the fundamental insights into rheology and printing parameters including the following in the introduction and the "Resin formulation and photocuring" section:

"This unprecedented approach demands a unique convergence of polymerization processes and organic catalysis together with the required rheology and photo-reactivity required for efficient vat photopolymerization...PBGs have received significantly less attention to free radical counterparts, but further fundamental understanding will provide much needed diversity to the range of polymeric precursors for vat photopolymerization..."

"A sharp photo-hardening profile is crucial for efficient VP process. By the choice of the TBD-HBPh₄:ITX photocatalyst system, this was achieved under mild UV-A irradiation intensities, which are required for its application in commercially available VP printers... Increasing the irradiance intensity to 200 mW/cm² (Figure S1b) resulted in a substantially faster rise of the storage modulus related to the faster activation of the PBG, although the use of such high intensities will limit the application of the system in conventional 3D printers. Our reported system differs from previously reported PBG-catalyzed PSU polymerization, as fast photocuring profiles were obtained using UV-A medium irradiation in contrast to high intensity Hg-Xe lamps, which are not suitable for the manufacturing technique.⁴⁴"

Besides, the reviewer commented the work of Zhao et al. (Reprintable polymers for digital light processing 3D printing. *Advanced Functional Materials*, 31(9), 2007173. In this work authors printed linear polymeric structures which were soluble in the system and therefore no chemical recycling was needed. Therefore, the approach used in the aforementioned work is not comparable to the one presented here. On the other side, the work reported by Wang et al. (Recyclable thermosetting polymers for digital light processing 3D printing. *Materials & Design*, 197, 109189 present an example of dual curing photoresist which are crosslinked in a first step by free radical photopolymerization of acrylates and a second thermal treatment to trigger the epoxy curing which induces a certain degree of copolymerization between the acrylate and epoxy networks. For the depolymerization a transesterification procedure was used by the addition of large excess of ethylene glycol (EG) and TBD at 180 °C and 8 h and then the excess of EG was removed by heating at 200 °C. In addition, in the reformulated photoresist, a 30 wt% of recycled residual mixture mixed with 70 wt% of fresh resin in each cycle. Beside, the thermal properties of the recycled photoresist formulations are substantially different since the nature of the recycled resin is notably altered. Therefore in that work, an elegant strategy to reuse previously printed parts was used although partial depolymerization of the objects was achieved since the C-C bond backbones generated during the printing

process were not recycled. In addition, the chemical depolymerization of the printed structures into liquid residue was achieved although the depolymerization products were not suitable for their reprint due to the absence of photocurable groups. This is derived from the difference between the photoactive chemistry and the transesterification reaction used for the chemical depolymerization step.

In our work, for the best of our knowledge we introduce a novel paradigm of 3D printable resins where the crosslinking points formed during the printing process (thiourethane) are the dynamic ones, substantially enhancing the circularity of the process by the built-in circular approach. These dynamic bonds can undergo rational depolymerization into thiol-terminated oligomers which are readily photoactive by the incorporation of fresh isocyanate into the system. This concept induces a more sustainable approach from our point of view.

We agree with the reviewer that maybe this was not enough emphasized in the current version and therefore we have modified the introduction discussion emphasizing previous works containing DCBs in for VP, as well as discussed previous approaches for reprintable formulations. For clarity, a Scheme (Scheme 1) has been also introduced to highlight the differences between the current and previous approaches on reprintable systems. Hopefully now the reviewer is more convinced about the novelty and impact of the work.

The new introduction reads as follows:

DCBs in VP has been explored in recent years due to the versatility offered by dynamic bonds in the printed structure. In this regard, printed objects with reshaping, self-healing and/or reprocessing possibilities are reported in recent reviews.^{29,30} The impact of DCBs in VP has significantly received less attention for reprintability possibilities of 3D printable materials. Zhao and coworkers reported a reprintable system based on the photopolymerization of a monofunctional acrylate to form linear polymers, which rapidly transform from liquid to 3D objects of linear polymers with high resolution.³¹ The printed polymers were subsequently dissolvable in the neat monomer to afford a reprintable resin. This approach addresses the recyclable limitations of VP printed thermosets although provides inferior high performance properties in terms of solvent resistance or thermo-mechanical property of thermosets (Scheme 1a). Wang and coworkers also recently reported a 3D printable recyclable two-stage curing thermoset system composed by photoactive acrylate and epoxy hybrid resin.³² The proposed mechanism follows traditional acrylate photocuring followed by a thermal treatment to initiate the epoxy polymerization together with a partial copolymerization of the epoxy-acrylate system. The subsequent thermosets were depolymerized using excess ethylene glycol to promote depolymerization via transesterification to afford a viscous liquid residue. However, the depolymerization residue was reformulated with fresh photopolymerizable resin to afford printable objects as no photoactive group remained available after the depolymerization process. Thus, the depolymerized oligomers did not bear photocurable groups and therefore, new acrylate sites were added after each cycle to perform the photopolymerization (Scheme 1b). Despite this emerging attention in the literature, the development of a more sustainable composition together with a sustainable process remains as an attractive opportunity for investigation.

Previous works

a) 3D printing of reprintable thermoplastics

b) Partial depolymerization acrylate-epoxy double networks

Current work

c) Direct photocuring of dynamic thiourethane bonds

Scheme 1: Recyclable 3D printable systems. a) Photoprinting of linear thermoplastics, which are soluble in the monomer. b) Photoprinting of dual systems containing photocurable monomers and heat curable monomers. Further depolymerization by dynamic covalent bond exchange to yield liquid residue and subsequent photocuring by addition of fresh resin. c) Loop-printing process where DCB are printed using photobase generators, which are further depolymerized using an excess of initial thiol monomer to yield functional oligomers which are reprintable by addition of isocyanate functional monomer.

Inspired from earlier literature using DCBs, our research delivers photocurable resins with circularity built into their performance using commercially available monomers. We envisioned that to afford a more circular approach for the preparation of photoprintable resins with retained properties, the DCB allowing the depolymerization must be the bond generated during the printing process. This unprecedented process allows to produce resin formulations and therefore photocured thermosets with retained properties to the pristine materials. Thus, conventional acrylate photopolymerization strategies are not suitable due to the non-dynamic nature of the formed C-C bonds of the thermoset backbone and therefore additional phototriggered strategies are required for the printing process (Scheme 1c).

Comment: The authors specifically choose a very loosely crosslinked system. The glass transition is around $-30\text{ }^{\circ}\text{C}$. How would the crosslinking density impact the materials in the depolymerization process? What is the mechanism of depolymerization process? Bulk or surface degradation? And what is the impact of the solvent (in this case, acetone) on the depolymerization kinetics? Would other solvents work for the depolymerization process?

Response: We thank the reviewer for the comment in this regard. In response to this comment, we have performed the photocuring of a PSU networks with higher crosslinking density by the mixture of IPDI and TMPTMP together with the reported photocatalyst system. The obtained PSU IPDI network showed a T_g at 58 °C. (See Figure S3). The depolymerization of this network was performed in the same conditions reported in the manuscript observing a total dissolution of the network after 8 h at room temperature. The resulting structures were analyzed by SEC and suggest the formation of the oligomeric thiol terminated structures. We have added two sentences in the manuscript as well as the DSC and SEC Figures in the SI document.

“The versatility of the process was also shown by forming PSU networks bearing IPDI as the di-isocyanate, which yielded a printed object with higher crosslinking density and a T_g of 58 °C. (see Figure S3).

“It should be mentioned that PSU networks composed of IPDI and triSH were also susceptible to depolymerization under identical conditions although longer times were needed, resulting in oligomeric thiol terminated structures as shown by SEC (See Figure S12). The depolymerization process was versatile for diverse compositions, thus enabling recyclability of printed objects of varied thermomechanical properties.”

Besides, depolymerization kinetics has been performed and characterized by ^1H NMR for the system containing **PSU1** with the addition of 25 mol % of toluene as internal reference. Results show a linear relationship between the depolymerization product formation and time, showing a complete conversion to the oligomeric products within 60 min (See Figure S10). These results suggest a surface depolymerization of the network since no induction time related to swelling of the material was observed. The depolymerization seems to be selective to different solvents. The depolymerization in diethyl ether did not work at the reported conditions suggesting the great impact of the solvent in this depolymerization procedure. Which might be related to the swelling capacity of the solvent. However, this has not been further investigated as it is out of scope of this work. We just wanted to show that these 3D objects could be easily recovered but we are aware that further optimization will be required to find the best conditions for depolymerization.

Now the depolymerization kinetics are described in the text as follows:

“Depolymerization kinetics showed a linear dependance of the depolymerization product formation and time suggesting surface depolymerization due to the absence of induction time (See Figure S10). “

Comment: Fig 3e, for the healing mechanism, the authors should comment on why it is a dissociative mechanism versus an associative mechanism. An FTIR or NMR spectrum showing intermediate thiol and isocyanate peaks at elevated temperatures should be provided to support this argument.

Response: We thank the reviewer for his comment in this regard. To verify the dissociative nature of the reported system, FTIR measurements have been carried out at reprocessing temperature (120 °C). Time evolution experiments clearly show an increase of the isocyanate stretching band supporting the dissociative mechanism (See Figure S9). The low intensity of the thiol band in the FTIR spectra did not allow the determination of thiol formation during the thermal process. A sentence has been included in the document:

“Most importantly, the dissociative dynamic mechanism of the thiourethane network was confirmed with FTIR spectroscopic studies at 120 °C where an increase of the transmittance of the isocyanate stretching band over time was readily observed, and where a maximum of the band was achieved after 68 min heating (See Figure S9).”

Comment: The authors show the printability of the resin without solvent. However, the resolution in Fig S6 looks far below the "good resolution" standard for DLP and two-photon lithography. The authors should also command the impact of the solvent on the mechanical properties. The tensile properties printed with and without acetone should be provided.

Response: We thank the reviewer for his comment regarding printing resolution. It is true that the resolution of printed structures without the use of acetone is substantially lower, however for low height 3D objects (Figure S6 left) equal resolution to the objects printed with the diluent can be observed. This might be due to the catalyst migration promoting lower degree of resolution for long prints and therefore limiting the scale of printable objects without the use of acetone. Of course, this is the first study and we believe that with the proper catalyst and resin this may be improved in future. We have added a sentence accordingly:

“...for low height objects (Figure S6). For higher objects ($h > 3$ mm) lower degree of resolution was achieved compared to for the analog of the resin containing the diluent...”

Discussion regarding the mechanical properties of samples printed without acetone is address together with the comment related to mechanical characterization of recycled samples.

Comment: The resins contain free isocyanate, a highly toxic and unstable species. Therefore, the authors should comment on the limitation of the resins.

Response: It is true that isocyanates have really bad reputation and for the use of isocyanate special training must be provided to the people preparing the 3D objects. We have to say that the number of isocyanates in these resins is reduced as we are using oligomeric structures, but we agree with the reviewer that ideally this concept should be implemented in a greener system. However, the current alternative to isocyanates are acrylate-based resins that in many cases are also toxic (a lot of concerns are coming with acrylics). Therefore, we have added a sentence for general users to be aware of this issue:

“Safety equipment must be used due to the use of thiols and isocyanate terminated prepolymer, which have shown some health concerns.”

About the stability we have shown that these resins are stable in the current conditions, of course you have to be careful with humidity but in general we have stored the prepolymer in the fridge for some time and we did not observe any change in viscosity. A sentence has been added:

“Indeed, in spite of the inherent reactivity of isocyanates these formulations could be stored at low temperature and dry environments up to a month without suffering any change in the resin.”

Comment: For Fig S13, the toughness of the material reduced dramatically with the recycled materials. For example, the 2nd recycled material shows $\sim 50\%$ reduction in the elongation/toughness. One unique advantage of the chemically recycling strategy is the ability to achieve similar mechanical properties. However, the results in the tensile data failed to provide the evidence for retaining the similar mechanical behavior.

Response: We thank the reviewer for his comment regarding the mechanical properties of the recycled materials. As pointed out, previous results showed a reduction of the mechanical performance after 2 recycling cycles. We believed that this performance lowering is derived from the lack of stoichiometry balance for the preparation of recycled resin formulations. As a way to overcome this issue, we have now implemented an iodometric titration approach to quantify the thiol functionality presence in the depolymerized oligomers and to be able to reformulate equimolar resins. By this approach, we have repeated mechanical testing studies and obtained retained mechanical properties. We have now included these observations in the text as follows:

“...after the quantification of the thiol content of the recycled oligomers by iodometric titration (Refer to Supporting Information), and irradiation with UV light centered at 365 nm and an irradiance of 20 mW/cm^2 for 30 mins, yielded the recycled materials which were characterized by FTIR spectroscopy (Figure 4b). Two cycles of this process resulted in materials with similar thermomechanical properties in terms of glass transition temperature (T_g) and storage modulus

between original and recycled PSU thermosets as shown in Figure 4d. Moreover, thermogravimetric analysis (TGA) and gel content (GC) experiments further confirmed the crosslinked nature of the recycled polymer network (See Figure S14 and Table S2). Interestingly, we found that in order to obtain recycled PSU materials with retained mechanical properties it is of major importance to maintain the stoichiometric balance between the thiol and isocyanate groups (due to the step-growth nature of the polymerization) and therefore quantify the thiol presence on the depolymerized oligomers. In this regard, uniaxial tensile testing of printed dumbbell-shaped samples showed mechanical property retainment, supporting the fact that chemical depolymerization allowed for the better property achievement after recycling cycles (Figure S17). The printing of **PSU1** without acetone yielded slightly lower mechanical performance compared to the **PSU1** material printed with acetone (Figure S18). This result was presumably attributed to lower monomer conversion due to gelation of the network and therefore this aspect would require further optimization and postcuring steps.”

Reviewers' Comments:

Reviewer #3:

Remarks to the Author:

Sardon and co-workers have provided a comprehensive response to the reviewers' recommendations. However, one concern remains: as the process necessitates adding more reactants with each cycle, the authors should discuss the proportion of recycled thiol oligomer originating from the pristine thiourethane network instead of the introduced thiol monomer.

Reviewer #3

Comment: Sardon and co-workers have provided a comprehensive response to the reviewers' recommendations. However, one concern remains: as the process necessitates adding more reactants with each cycle, the authors should discuss the proportion of recycled thiol oligomer originating from the pristine thiourethane network instead of the introduced thiol monomer.

Response: We thank the reviewer for their comment in this regard. As mentioned, each depolymerization cycle requires the addition of fresh triSH to obtain fully depolymerized oligomers. In this regard, 19.5 wt% of triSH needs to be added to PSU1 material for each depolymerization cycle (0.39 g of triSH for 2.00 g of PSU1). This discussion has been added in the manuscript which now can be read as follows:

“It should be mentioned that by the addition of 19.5 wt% of triSH in each depolymerization cycle PSU1 totally depolymerizes into oligomeric structures.”